# Adherence to antiretroviral therapy and its associated factors among children living with HIV in Eastern and Southern Africa: A systematic review and meta-analysis

Gebrie Getu Alemu ⓘ *, Bantie Getnet Yirsaw, Tigabu Kidie Tesfie ⓘ, Getaneh Awoke Yismaw, Habtamu Wagnew Abuhay, Meron Asmamaw Alemayehu, Muluken Chanie Agimas, Nebiyu Mekonnen Derseh ⓘ

Department of Epidemiology and Biostatistics, Institute of Public Health, College of Medicine and Health Sciences, University of Gondar, Gondar, Ethiopia

* gebryegetu27@gmail.com

**Data Availability Statement:** All data are in the manuscript and/or supporting information files.

## Abstract

### Background

Children living with HIV in low-income settings, such as in Eastern and Southern Africa, are at a high risk for poor adherence to antiretroviral therapy. However, various primary studies presented inconsistent and inconclusive estimates of adherence and its associated factors among children living with HIV in Eastern and Southern Africa. Therefore, we were aimed to determine the pooled prevalence of adherence and its associated factors, and to guide intervention efforts to support adherence, this comprehensive systematic review and meta-analysis was conducted.

### Methods

We have comprehensively searched PubMed, Google Scholar, EMBASE, Scopus, and Hinari databases for all primary studies. Unpublished studies were also searched manually and accessed from university repositories. Additional searches were conducted by examining the references in the included articles to further identify relevant studies. Data were extracted and analyzed using Microsoft Excel spreadsheet and STATA version 17 software, respectively. A random-effects DerSimonian-Laird model was used to compute the pooled prevalence of adherence to antiretroviral therapy among children living in Eastern and Southern Africa. We have used Cochran's Q test ($\chi2$) and Higgins $I^2$ statistics to identify heterogeneity. Subgroup and sensitivity analysis were conducted to investigate the potential sources of heterogeneity. Publication bias was assessed by the funnel plot and Egger's test. An association was expressed through the pooled adjusted odds ratio and statistical significance was considered at a p-value < 0.05.

**Funding:** The author(s) received no specific funding for this work.

**Competing interests:** The authors declare that we have no conflict of interest with anyone with respect to the research, authorship, and/or publication of this article.

**Abbreviations:** HAART, Highly Active Antiretroviral Treatment; AIDS, Acquired Immune Deficiency Syndrome; ART, Antiretroviral Therapy; AOR, Adjusted Odds Ratio; CD4, Cluster of Differentiation; CI, Confidence Interval; ESA, Eastern and Southern Africa; HIV, Human Immunodeficiency Virus; JBI, Joanna Briggs Institute; PRISMA, Preferred Item for Systematic Review and Meta-Analysis; SSA, Sub-Saharan Africa; WHO, World Health Organization.

## Results

This meta-analysis combined the effect estimates of 29 primary studies with 7414 study participants. The pooled prevalence of adherence to antiretroviral therapy among children living in Eastern and Southern Africa was 76.2% (95% CI: 71.4, 81.1) [$I^2$ = 97.06%, P < 0.001 and Q test ($\chi^2$) = 953.83, p-value < 0.001]. Being a biological caregiver [AOR = 1.93 (95% CI: 1.34, 2.73)], receiving first-line antiretroviral treatment [AOR = 2.7 (95% CI: 1.39, 5.25)], and having social support [AOR = 1.88 (95% CI: 1.33, 2.66)] were significantly associated with adherence to antiretroviral therapy.

## Conclusion

The pooled prevalence of adherence to antiretroviral therapy among children living with HIV is low. Biological caregiver, first-line antiretroviral treatment, and social support were factors associated with adherence to ART among children living with HIV. Therefore, healthcare providers, adherence counselors, supporters, as well as governmental and non-governmental organizations, should emphasize a multi-component intervention approach to address the multifaceted challenges associated with adherence to ART, thereby improving counseling efforts to enhance adherence. Moreover, clinicians should prioritize the selection and utilization of regimens for individuals in this age group on robust first-line options.

## Introduction

The Human Immunodeficiency Virus (HIV) remains an important public health problem worldwide [1]. The international commitment to ending the Acquired immunodeficiency syndrome (AIDS) epidemic by 2030 places HIV prevention at the core of the response. Reducing HIV infections in the Eastern and Southern African (ESA) region is essential to reaching this challenging goal because the illness still disproportionately affects people in this region [2]. Globally, as of 2023, there were 1.4 million children (< 15 years) living with HIV. Of this 120,000 of them were acquiring new HIV infections and 76,000 children were dying from HIV-related causes [3]. Globally, in 2022, only 57% of children aged < 15 years had access to antiretroviral therapy (ART) [4]. In children, the prevalence of adherence to ART ranged from 43% [5] to 96% [6] in ESA.

As of 2023, globally, 39.9 million people were living with HIV. In the same year, about 1.3 million people became newly infected with HIV, and 630,000 people died from AIDS-related illnesses. There were 26 million people living with HIV, 760,000 new HIV infections, and about 390,000 people died from AIDS-related illnesses in Africa at the end of 2023. Of this, 20.5 million people living with HIV, 440,000 new HIV infections, and 240,000 AIDS-related deaths occurred in ESA [3]. This makes HIV/AIDS the highest burden and a major public health concern in ESA. Consequently, Adherence to Highly Active Antiretroviral Therapy (HAART) coverage is increasing worldwide; only 29.8 million (76%) of all people living with HIV were taking HAART at the end of December 2022 [4].

Antiretroviral therapy comprises a combination of antiretroviral drugs aimed at reducing the viral load in the body to minimal levels, increasing the patient's immune system by increasing the number of Cluster of Differentiation 4 (CD4) cells, determining treatment efficacy, averting viral resistance, and improving the quality of life for individuals afflicted with HIV/AIDS [7].

Effective management of HIV/AIDS relies on adherence to ART, ensuring optimal control of viral load and CD4 levels while preventing further complications. Sustained adherence is essential for treatment success, though maintaining compliance with ART often poses unique challenges, requiring commitment from both the patient and healthcare provider [8]. Adherence to HAART is the backbone approach in reducing morbidity and mortality among children infected with HIV [9]. Adherence to ART is the robust predictor of successful treatment of HIV among children [10]. Providing ART was one of the major therapeutic obstacles, but maintaining children with HIV/AIDS at the appropriate degree of adherence is crucial to the effectiveness of their therapy [11]. Antiretroviral therapy is now more widely available, but its effectiveness still depends on adherence to a lifelong antiretroviral regimen [12].

Previous research evidence states that socio-demographic, clinical, and treatment-related factors like age of the child [9, 13–18], sex of the child [13, 16, 19], residence [19–21], educational level of care givers [5, 6, 14, 15, 22–24], knowledge of caregiver on adherence [9, 11, 18], disclosure of HIV status to the children [14, 18, 19, 21, 23, 25, 26], distance from health facility [9, 19, 26], relationship of caregivers with healthcare providers [18, 19, 27], treatment stage [9, 16, 19, 25], ART side effect [27], relationship of caregivers with the child [21, 22], having social support [22, 27], hemoglobin level [25], CD4 counts [28] and WHO disease stage [9, 15, 25, 29] were significant predictors of adherence to ART.

In order to achieve the United Nation's Sustainable Development Goals and the UNAIDS "95-95-95" targets, it is essential to prioritize enhancing adherence interventions among individuals living with HIV/AIDS, particularly within the younger group [30]. Human Immunodeficiency Virus and Acquired Immune Deficiency Syndrome is the highest burden and major public health problem in ESA with poor level of adherence to ART. In children various primary studies exhibited inconsistent and inconclusive estimates on adherence to ART and demonstrated epidemiological variations ranging from 43% [5] to 96% [6] in ESA. Nevertheless, there is currently no representative information regarding the pooled prevalence of adherence to ART among children living with HIV in this HIV/AIDS-heavy region. Understanding the pooled levels and associated factors of adherence to ART is crucial in designing interventions and planning health services to improve adherence and health outcomes of ART. Therefore, the aim of this comprehensive systematic review and meta-analysis was to estimate the pooled prevalence of adherence and identify factors associated with adherence to ART among children living with HIV in Eastern and Southern Africa.

## Methods

### Reporting and registration of protocol

A 27-item Preferred Reporting Items for Systematic Reviews and Meta-Analyses (PRISMA 2020) checklist [31] was used to report the results of this systematic review and meta-analysis (**S1 Table**). The study protocol for this review has been registered in the International Prospective Register of Systematic Reviews (PROSPERO) with identification number CRD42024505507.

### Databases and search strategy

We have comprehensively searched PubMed, Google Scholar, EMBASE, Scopus, and Hinari through research4life and other manual searches for all available primary studies reporting adherence and its predictors in the ESA using the following search terms and phrases: "antiretroviral agents"; "antiretroviral therapy, highly active"; "adherence to antiretroviral therapy"; adherence; "non-adherence"; HIV; "HIV Infections"; "acquired immunodeficiency syndrome"; "Human Immunodeficiency Virus Infection"; Child; Children; Infant; Botswana; Burundi;

Comoros; Djibouti; Eritrea; Eswatini; Ethiopia; Kenya; Lesotho; Madagascar; Malawi; Mauritius; Mozambique; Namibia; Rwanda; Seychelles; Somalia; "South Africa"; "South Sudan"; Swaziland; Tanzania; Uganda; Zambia; Zimbabwe; "associated factors"; risk factors"; predictors. Search strings were implemented using "AND" and "OR" Boolean operators. The search was further limited to articles published in English. We employed the Population, Exposure, Context, and Outcome (PECO) mnemonic framework for systematic reviews to organize the search by using Boolean operator combinations. Additional searches were conducted manually by examining the references of the included articles, along with further investigations of citations to find out additional studies that reported medication adherence in children within ESA that were not initially found through electronic database searches. The search was conducted from February 1 to February 26, 2024. A separate file with the search details was supplied (S2 Table).

## Eligibility criteria

All articles that were conducted in Eastern and Southern Africa countries, written in English, published in the years between January 2015 and February 26, 2024, designed as an observational study (cross-sectional, prospective, or retrospective cohort), and experimental studies that explicitly measured adherence to ART among children aged < 15 years on ART and/or at least one associated factor for adherence to ART were eligible for inclusion in this study. Studies without abstracts, full texts, systematic reviews and meta-analyses, qualitative studies, studies that sampled adolescents and adults, and studies that did not pass our quality screening were excluded from the review. All the retrieved citations of identified documents were exported to EndNote version 20 reference managers to remove duplicate studies and screened for relevance. Screening was carried out at three levels: title, abstract, and full text. Initially, two reviewers (GGA and BGY) independently screened the titles and abstracts to consider the articles in the full-text review. Full texts of the potentially eligible abstracts were downloaded, read, and subsequently screened to ascertain their relevance with respect to the inclusion criteria.

## Quality appraisal and risk of bias assessment of the included studies

The Joanna Briggs Institute's (JBI) critical appraisal tools consisted of 8 components for analytical cross-sectional studies, 11 components for cohort studies [32] and the Cochrane Handbook for Systematic Reviews of Interventions comprises eight components for experimental studies [33] were employed to evaluate the quality of the included studies. Three authors (GGA, MAA, and MCA) independently assessed the methodological quality of each study. Disagreements in the assessment and scoring were solved through discussion and consensus. The interpretation of risk of bias was conducted as follows: for analytic cross-sectional studies, articles scoring 7–8, 4–6, and 0–3 points were classified as having low, moderate, and high risk of bias, respectively. For cohort studies, scoring 9–11, 5–8, and 0–4 points indicated a low, moderate, and high risk of bias, respectively [34]. For an experimental study, if an article scored 7–8, 4–6, and 0–3 points was classified as low, unclear, and high risk of bias [33]. Studies with a low to moderate risk of bias after being evaluated against these criteria were included in this systematic review and meta-analysis (S3 Table).

## Data extraction

Data were extracted by two independent reviewers (GGA and TKT) using a structured Microsoft Excel spreadsheet. The phase was repeated each time differences in the extracted data were observed. The third reviewer (HWA) was involved while the differences between the data

extractors continued. The name of the first author, year of publication, study setting, sampling method, study design, sample size, response rate, adherence measurement tools, and exposures to the outcomes of the included primary studies were extracted.

## Outcome measures

The primary outcome of interest was the pooled prevalence of adherence to ART which was determined by dividing the total number of adherences to ART by the total number of study participants multiplied by one hundred and the secondary outcome of interest was the factors associated with adherence to ART with variables identified as risk factors for ART adherence in at least two primary studies among children living in ESA.

**Operational definitions of variables.** **Adherence to ART** refers to taking $\geq$ 95% of prescribed ART medications by children living with HIV/AIDS based on WHO guidelines, and those HIV-positive children on ART experienced fair or poor adherence (drug adherence of < 95%) were considered **non-adherent to ART** [35, 36].

Although photographs or videos linked to an individual's recorded data are not included in the study, there is no conflict about consent to publish other data obtained.

## Ethical approval and consent to participate

Not applicable because there were no primary data collected.

## Results

### Search outcome and study characteristics

The search strategy retrieved a total of 1714 studies from various databases, including PubMed (n = 760), Google Scholar (n = 617), EMBASE (n = 148), Scopus (n = 141), and Hinari through research4life (n = 48). Of these, 49 duplicated studies were identified and subsequently removed. We excluded 1611 irrelevant studies after screening titles and abstracts, and 54 studies were selected for eligibility (full text review). After conducting full text reviews, 34 studies were removed with reasons: (n = 29) were due to different target groups (target populations were not children <15 years), (n = 4) were qualitative studies, (n = 1) was the outcomes of interest not well defined and 20 studies were included in the review. Among the manually searched studies (n = 31), 10 studies were not retrieved, and 21 reports were assessed for eligibility. After conducting full text reviews, 12 studies were removed with the following reasons: (n = 9) were due to different target groups (target populations were not children <15 years), (n = 2) were qualitative studies, (n = 1) was the outcome of interest not well defined and 9 studies were included in the review. Overall we have excluded 1636 studies with reason (**S4 Table**). Finally, 29 studies met our inclusion criteria to determine the pooled prevalence of adherence to ART among children and its associated factors. We followed the PRISMA 2020 checklist flowchart to illustrate the selection process, detailing the progression from the initially identified records to the studies ultimately included in the analysis (**Fig 1**).

Among the twenty-nine finally included studies [5, 6, 11–29, 37–44]: twelve studies were conducted in Ethiopia [6, 11, 12, 15, 16, 18–21, 25, 37], seven studies were conducted in Kenya [23, 26, 28, 38, 40, 41, 44], three studies were done in South Africa [13, 42, 43], three studies were done in Tanzania [24, 29, 39], another three studies were done in Uganda [5, 17, 22], and one study was done in South Sudan [27]. Twenty six studies were conducted in Eastern Africa [5, 6, 11, 12, 14–29, 37–41, 44], and the rest three were done in Southern Africa [13, 42, 43]. Regarding study design, twenty-one studies were conducted using cross-sectional, six studies were employed with cohorts (3 prospective and 3 retrospective); one study was done using

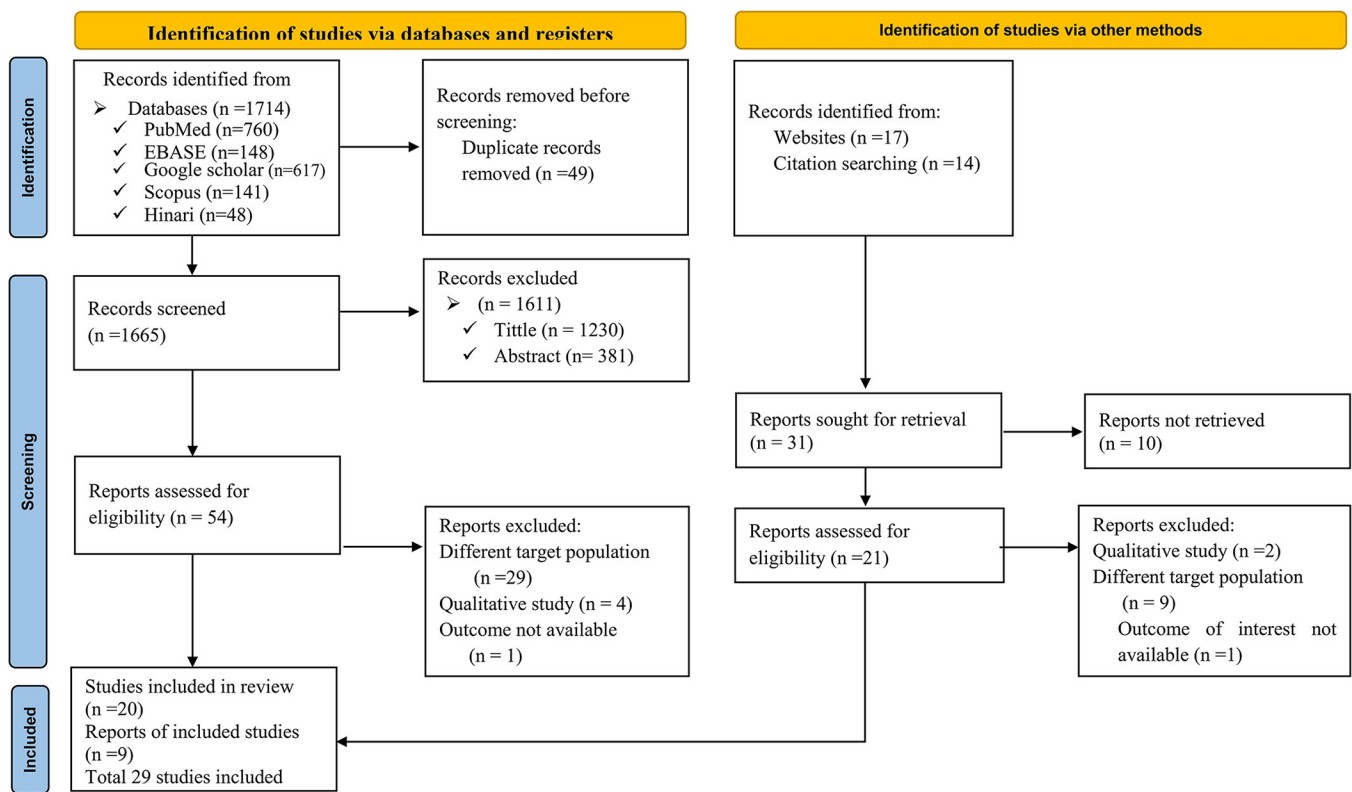

**Fig 1. Preferred Reporting Items of Systematic Reviews and Meta-Analysis flow diagram showing search results and study selections for meta-analysis on adherence to ART among children in Eastern and Southern Africa from 2015 to 2024 [31].**

experimental design; and the remaining one article study design was not recorded. Moreover, regarding to adherence measurement; twenty six studies were done with care giver self-report [5, 6, 11–25, 27–29, 37, 38, 40, 41, 43, 44], two studies were done with pill count [26, 42] and the other one study was done with drug plasma concentration [39]. Furthermore, the total sample size of the included studies was 7414, where the smallest sample size was 78 in South Africa and the largest sample size was 860 in Kenya with response rate range from 90.7 to 100% (**Table 1**).

## Data synthesis and statistical analysis

The extracted data were exported to STATA version 17 software for statistical analysis. Graphical presentations were employed to present a summary of pooled estimates through forest plots, along with visual assessments of publication bias. The characteristics of the primary studies included were summarized and displayed in the text and table. The summary table briefly presents key information such as author names, publication years, and study countries, study designs, sampling methods, and sample sizes, quality of studies, response rate, adherence measurements methods and major findings. The pooled prevalence of adherence to ART among HIV-infected children was obtained from twenty-nine included primary studies [5, 6, 11–29, 37–44], whereas the data regarding the determinants of adherence to ART were obtained from the twenty-five studies [5, 11–25, 27–29, 37–39, 41–43]. We employed the Cochran's Q test ($\chi$2) and Higgins $I^2$ statistics to identify heterogeneity. The Cochran's Q test ($\chi^2$) with a p-value < 0.10 indicates statistically significant heterogeneity [45]. An $I^2$ statistic of

**Table 1. General characteristics of included studies on adherence to ART in Eastern and Southern Africa from 2015 to 2024.**

| SNo | Author [Year] | Study country | Region | Study design | Sample size | Response Rate | method of adherence measurement | Prevalence (95% CI) | Quality |
|-----|---------------|---------------|--------|--------------|-------------|---------------|--------------------------------|---------------------|---------|
| 1 | Alemayehu B [2023] [19] | Ethiopia | East Africa | CS | 237 | 92% | Care giver self-report | 78.1(72.8, 83.3) | Low risk |
| 2 | Biru M [2017] [37] | Ethiopia | East Africa | Cohort | 306 | 99% | Care giver self-report | 92.8(89.9, 95.7) | Low Risk |
| 3 | Ebrahim H [2017] [13] | South Africa | South Africa | NR | 151 | 98.05% | Care giver self-report | 94.0(90.3, 97.8) | Moderate risk |
| 4 | Echiru A [2017] [22] | Uganda | East Africa | CS | 188 | 100% | Care giver self-report | 59.6(52.6, 66.6) | Moderate risk |
| 5 | Feyera B [2016] [14] | Ethiopia | East Africa | CS | 380 | 95% | Care giver self-report | 93.4(90.9, 95.9) | Moderate risk |
| 6 | Feyissa A [2017] [15] | Ethiopia | East Africa | CS | 120 | 93.75% | Care giver self-report | 64.2(55.6, 72.7) | Low risk |
| 7 | GebreEyesus F [2021] [12] | Ethiopia | East Africa | Cohort | 363 | 95.3% | Care giver self-report | 78.2(74.0, 82.5) | Moderate risk |
| 8 | Gemechu GB [2023] [20] | Ethiopia | East Africa | CS | 282 | 97.2% | Care giver self-report | 87.2(83.3, 91.1) | Moderate risk |
| 9 | Gultie T [2015] [16] | Ethiopia | East Africa | CS | 226 | 100% | Care giver self-report | 90.7(86.9, 94.5) | Low risk |
| 10 | Gutema EA [2021] [6] | Ethiopia | East Africa | CS | 80 | 100% | Care giver self-report | 96.3 (92.1,100.4) | Low risk |
| 11 | Guyo TG [2023] [25] | Ethiopia | East Africa | Cohort | 323 | 100% | Care giver self-report | 69.7(64.6, 74.7) | Moderate risk |
| 12 | Mugambi D [2015] [38] | Kenya | East Africa | CS | 214 | 100% | Care giver self-report | 87.4(82.9, 91.8) | Moderate risk |
| 13 | Mugusi SF [2019] [39] | Tanzania | East Africa | CS | 216 | 100% | DPC | 72.2(66.2, 78.2) | Moderate risk |
| 14 | Mukami C [2022] [40] | Kenya | East Africa | CS | 173 | 100% | Care giver self-report | 54.9(47.5, 62.3) | Low risk |
| 15 | Musovya J [2020] [41] | Kenya | East Africa | CS | 195 | 100% | Care giver self-report | 47.2(40.2, 54.2) | Low risk |
| 16 | Mussa FM [2022] [29] | Tanzania | East Africa | CS | 333 | 100% | Care giver self-report | 59.8(54.5, 65.0) | Moderate risk |
| 17 | Mwiti PK [2023] [28] | Kenya | East Africa | Cohort | 221 | 100% | Care giver self-report | 75.6(69.9, 81.2) | Low risk |
| 18 | Opiyo R [2022] [23] | Kenya | East Africa | Exp | 860 | 100% | Care giver self-report | 72.6(69.6, 75.5) | Low risk |
| 19 | Smith C [2016] [42] | South Africa | South Africa | Cohort | 78 | 100% | Pill count | 87.2(79.8, 94.6) | Moderate risk |
| 20 | Ssanyu JN [2020] [5] | Uganda | East Arica | CS | 206 | 100% | Care giver self-report | 42.7(36.0, 49.5) | Moderate risk |
| 21 | Talam N [2015] [26] | Kenya | East Africa | CS | 230 | 100% | Pill count | 56.1(49.7, 62.5) | Low risk |
| 22 | Tesfahunegn TB [2023] [11] | Ethiopia | East Africa | CS | 250 | 98.03% | Care giver self-report | 84.8(80.3, 89.3) | Moderate risk |
| 23 | Tiruneh CM [2022] [21] | Ethiopia | East Africa | CS | 383 | 90.7% | Care giver self-report | 68.1(63.5, 72.8) | Low risk |
| 24 | Tong PD [2020] [27] | South Sudan | East Africa | CS | 126 | 100% | Care giver self-report | 69.8(61.8, 77.9) | Low risk |
| 25 | Urassa DP [2018] [24] | Tanzania | East Africa | CS | 423 | 100% | Care giver self-report | 68.6(64.2, 73.1) | Low risk |
| 26 | Van Elsland SL [2018] [43] | South Africa | South Africa | CS | 195 | 100% | Care giver self-report | 89.2(84.9, 93.6) | Moderate risk |

(*Continued*)

**Table 1.** (Continued)

| SNo | Author [Year] | Study country | Region | Study design | Sample size | Response Rate | method of adherence measurement | Prevalence (95% CI) | Quality |
|-----|---------------|---------------|--------|--------------|-------------|---------------|--------------------------------|---------------------|---------|
| 27 | Vreeman RC [2015] [44] | Kenya | East Africa | Cohort | 191 | 100% | Care giver self-report | 93.2(89.6, 96.8) | Moderate risk |
| 28 | Wadunde I [2018] [17] | Uganda | East Africa | CS | 153 | 100% | Care giver self-report | 79.1(72.6, 85.5) | Low risk |
| 29 | Zegeye S [2015] [18] | Ethiopia | East Africa | CS | 313 | 100% | Care giver self-report | 90.7(87.5, 93.9) | Low risk |

CS, Cross-Sectional; NR, Not Recorded; Exp, Experimental; Drug Plasma Concentration

Data were extracted by two independent reviewers Gebrie Getu Alemu and Tigabu Kidie Tesfie.

Data were extracted from March 20 to March 27, 2024

0%, 25%, 50%, and 75% corresponds to no, low, moderate, and high levels of heterogeneity, respectively [46]. An $I^2$ estimate of more than 50% indicated significant heterogeneity [47]. Because the $I^2$ test ($I^2$ = 97.06, P <0.001) and Q test ($\chi^2$) = 953.83 with a p-value < 0.001 of this study for heterogeneity showed significant differences between studies, a random-effects Der-Simonian-Laird model was used to compute the pooled prevalence of adherence to ART among children living in ESA. Potential sources of statistical heterogeneity were assessed using sensitivity, and subgroup analyses. A sensitivity analysis was conducted to identify the impact of a single study on the overall meta-analysis. Publication bias was assessed subjectively by examining the symmetry of the funnel plot and objectively by Egger's test with a p-value of < 0.001. The impact of the independent variables on the outcome variable and a measure of association reported at a 95% confidence interval were analyzed and presented using tables and a forest plot, and statistical significance was considered at a p-value < 0.05. Texts, tables, and figures were used to present the results.

## Pooled prevalence of adherence to ART among children living with HIV

The prevalence of adherence to ART among the study participants ranged from 42.7% (95% CI: 35.9, 49.5) to 96.3% (95% CI: 92.1, 100.4). The overall pooled prevalence of adherence to ART among children living with in ESA was 76.2% (95% CI: 71.4, 81.1) with significant heterogeneity ($I^2$ = 97.06%, P < 0.001 and Q test ($\chi^2$) = 953.83, p-value < 0.001) (**Fig 2**).

## Heterogeneity

The overall $I^2$ statistic in the forest plot reveals a significant level of heterogeneity ($I^2$ = 97.06, p < 0.001). Hence, subgroup and sensitivity analysis were conducted.

## Subgroup analysis by study year of publication

The pooled prevalence of adherence to ART in studies conducted before the year 2020 was 81.6% (95% CI: 76.2, 87.1, $I^2$ = 95.95, p < 0.001), which was higher than studies conducted in the year 2020 and later 70.6% (95% CI: 63.5, 77.7, $I^2$ = 96.63, p < 0.001 (**Fig 3**).

## Subgroup analysis by geographical area (country)

Subgroup analysis was conducted based on the geographical area (study country) in the ESA where primary studies were conducted. Thus, we observed geographical variations in the pooled prevalence of adherence to ART in this review. In random effects pooled subgroup meta-analysis, the prevalence of adherence to ART in children ranged from 60.5% (95% CI:

## Pooled prevalence of Adherence to ART among children in ESA

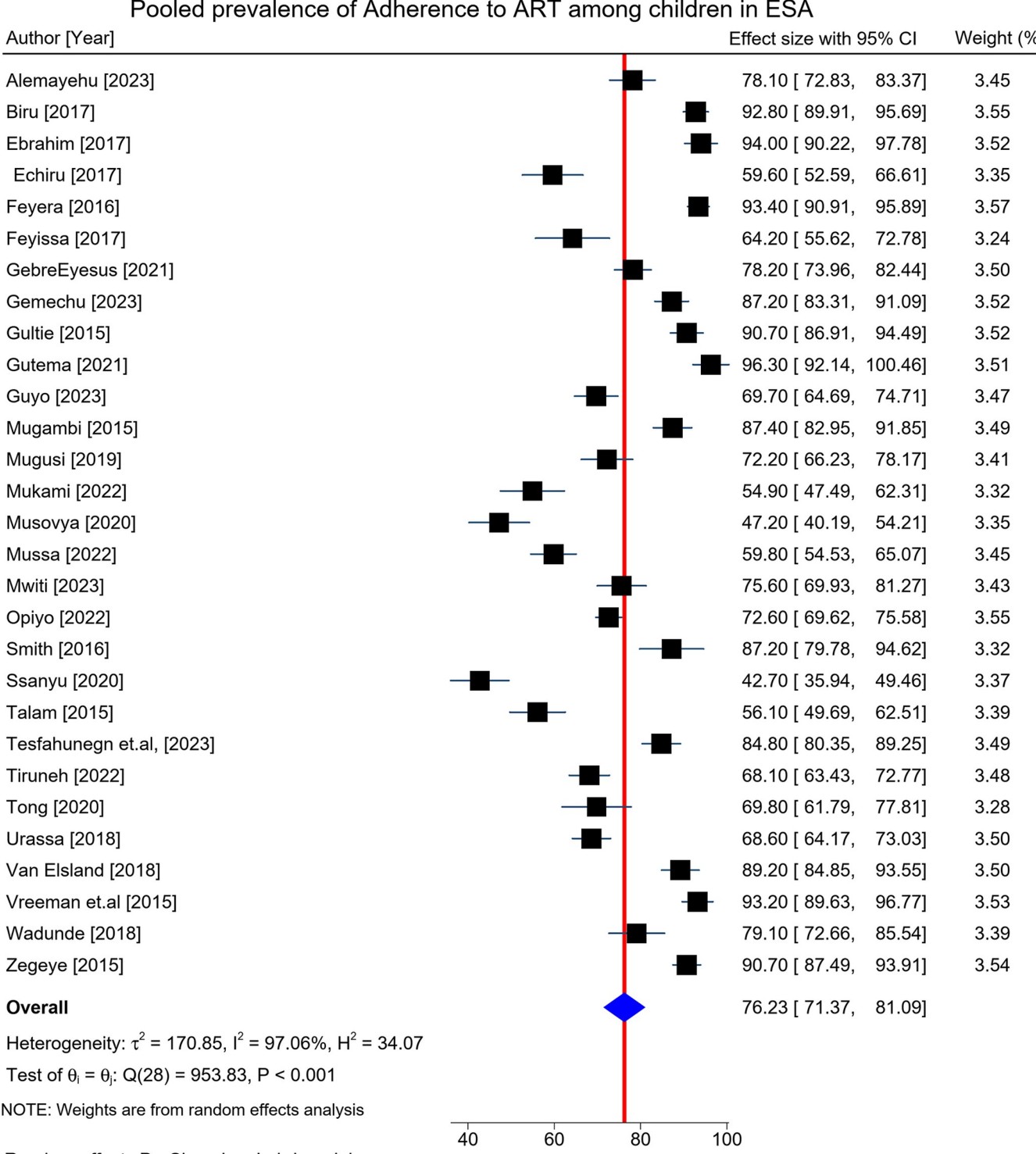

**Fig 2. A forest plot of 29 studies on the prevalence of adherence to ART among HIV-infected children in Eastern and Southern Africa from 2015 to 2024.**

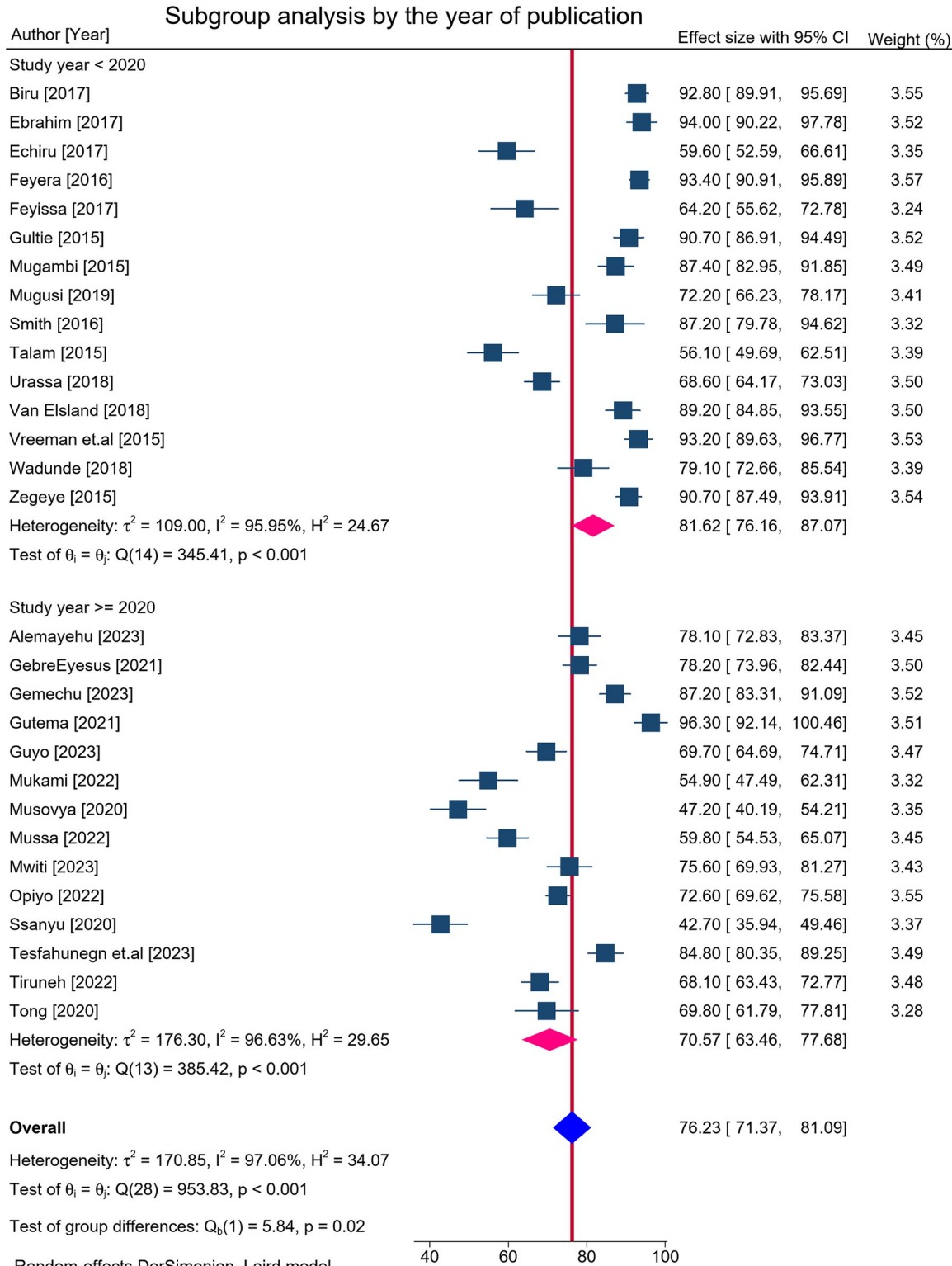

**Fig 3. Subgroup analysis by study year of publication among HIV-infected children with ART in Eastern and Southern Africa from 2015 to 2024.**

39.4, 81.5) in Uganda to 90.8% (95% CI: 86.8, 94.8) in South Africa, with various heterogeneities (**Fig 4**).

## Subgroup analysis by adherence measurements in the primary studies

The subgroup analysis by adherence measurements of ART among the primary studies showed deference in the pooled prevalence of adherence to ART. In random effects pooled subgroup meta-analysis, the prevalence of adherence among care giver self-report (reported questionnaire) and pill count were 76.7% (95% CI: 71.7, 81.8), and 71.6% (95% CI: 41.1, 102.1), respectively (**S1 Fig**).

## Sensitivity analysis

We conducted a sensitivity analysis to assess the robustness of the synthesized results and impact of a specific study on the overall meta-analysis findings. Sensitivity analysis identifies particular decisions or missing information that greatly influence the findings of the review. An individual study is suspected of excessive influence if the point estimate of its "omitted" analysis lies outside the confidence interval of the "combined" analysis. Furthermore, a study is excessively influential if its "omitted" meta-analytic estimate differs in significance relative to the "combined" analysis. Therefore, the forest plot of this review indicated that the estimate from an individual study aligns closely with the combined estimate, and no point estimate of its omitted analysis lies outside the confidence interval (CI) of the overall pooled estimate, suggesting the lack of a significant individual study effect on the overall pooled estimate. Hence, it has been demonstrated that a single study did not exert a substantial influence on the overall outcome of the meta-analysis (**Fig 5**).

## Publication bias

The inclusion of primary studies in the funnel plot suggested publication bias, as shown by the plot's asymmetry due to the unequal distribution of studies on its left and right sides (**Fig 6a**).

Furthermore, the p-values of Egger's regression test (p < 0.001) also indicated the presence of publication bias. Hence, we have conducted trim and fill analysis to manage the publication bias (**Fig 6B**).

## Factors associated with adherence to ART among children living with HIV in ESA

In this systematic review and meta-analysis, thirteen studies [5, 13, 14, 17–22, 29, 37–39] reported caregiver-child relationship was significantly associated with adherence to ART among children living with HIV. Being a biological caregiver was 1.93 times more likely to adhere to ART as compared with non-biological caregiver [AOR = 1.93 (95% CI: 1.34, 2.73)] (**Fig 7**).

The length of the segment and the midpoints on it indicated the OR of each included study with its corresponding 95% CI, and the diamond shape showed the combined AOR.

Moreover, three studies [16, 19, 27] reported first-line ART treatment was significantly associated with adherence to ART. Children on first-line ART treatments were 2.7 times more likely to adhere than those on second-line ART treatments [AOR = 2.7 (95% CI: 1.39, 5.25)]. Furthermore, the other three studies [21, 22, 28] reported social support was significantly associated with adherence to ART. Children who had social support were 1.88 times more likely to adhere to ART than those who did not have social support [AOR = 1.88 (95% CI: 1.33, 2.66)] (**Table 2**).

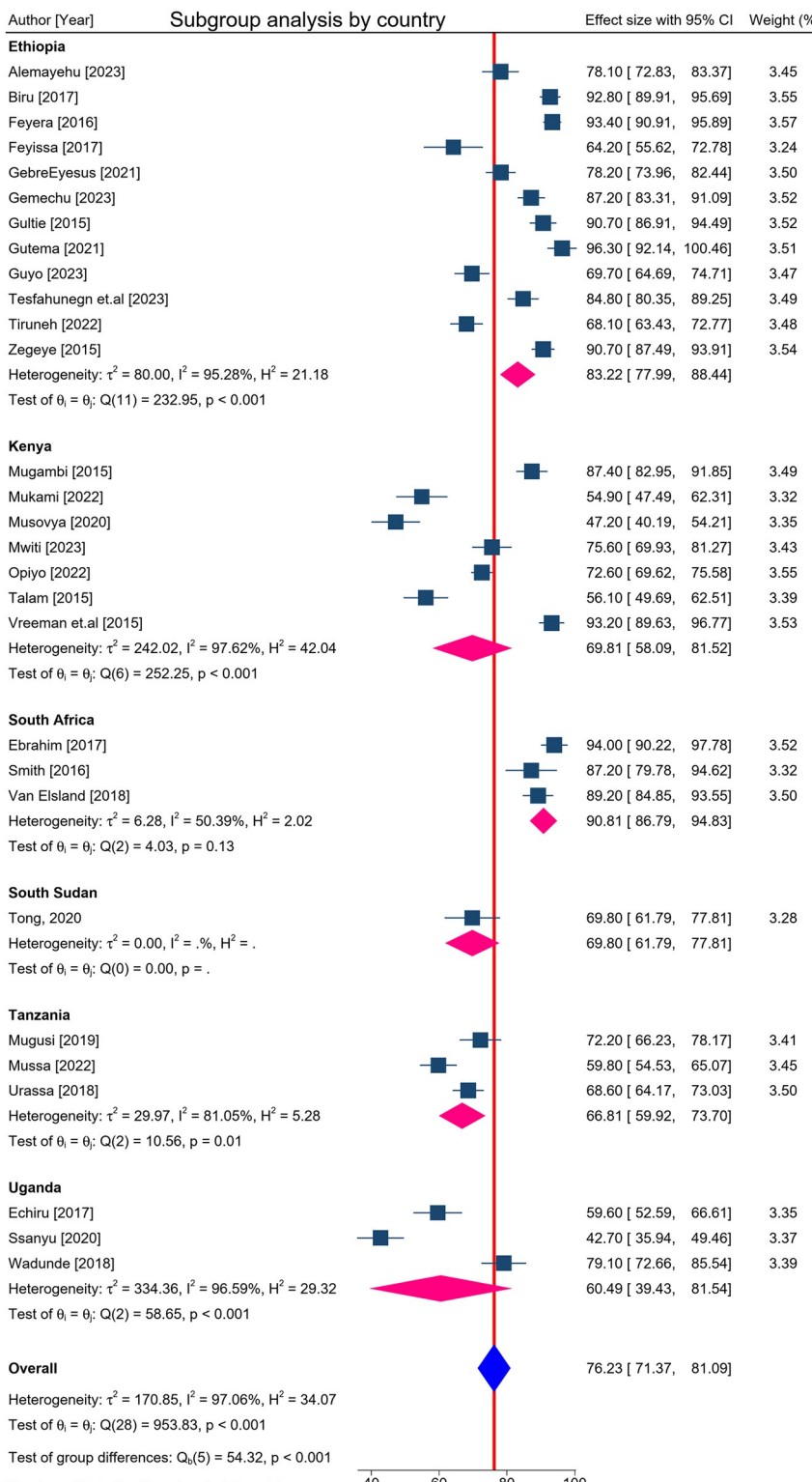

**Fig 4. Subgroup analysis by study country among HIV-infected children with ART in Eastern and Southern Africa from 2015 to 2024.**

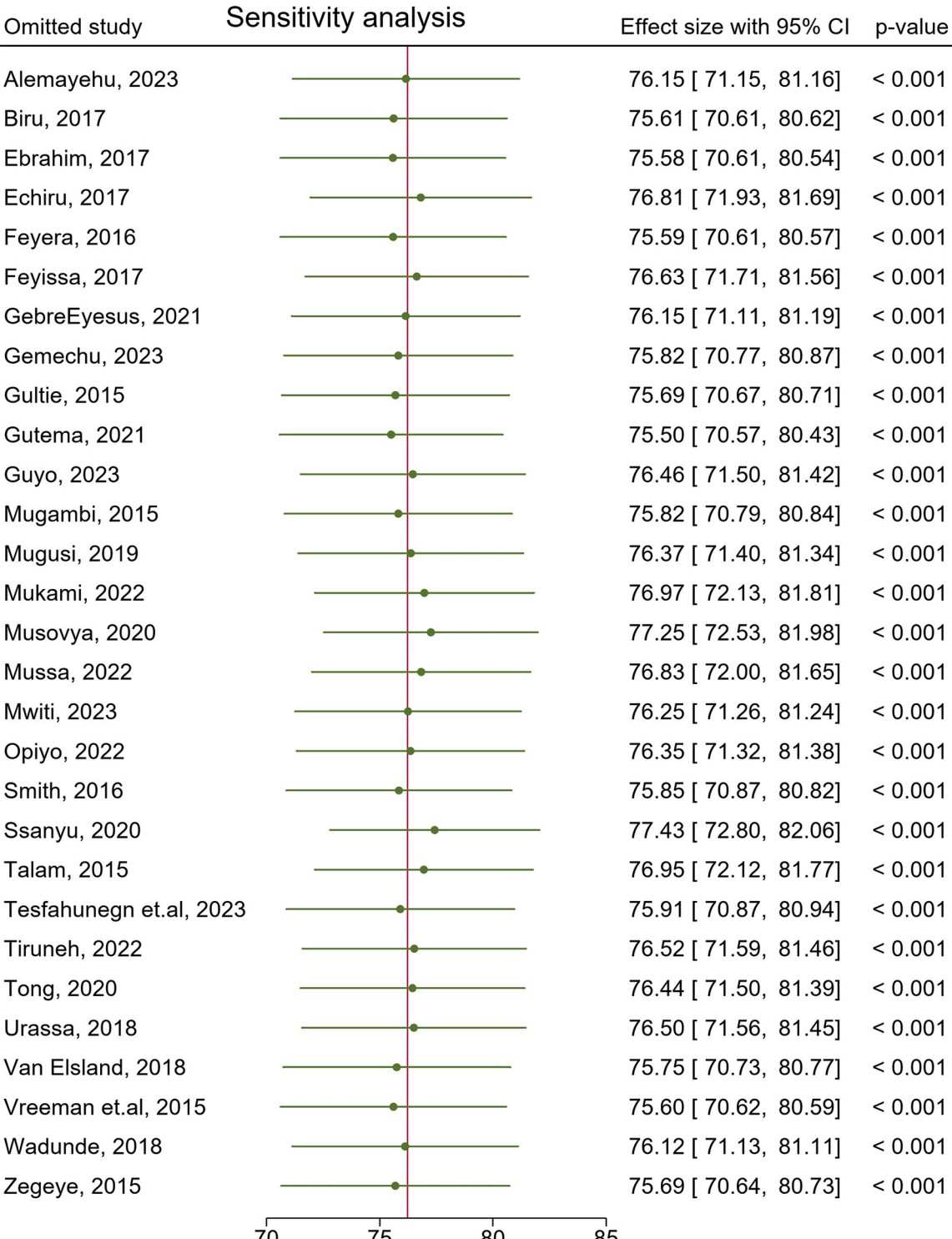

**Fig 5. Sensitivity analysis to assess the robustness of the synthesized results of adherence to ART among children with HIV in Eastern and Southern Africa from 2015 to 2024.**

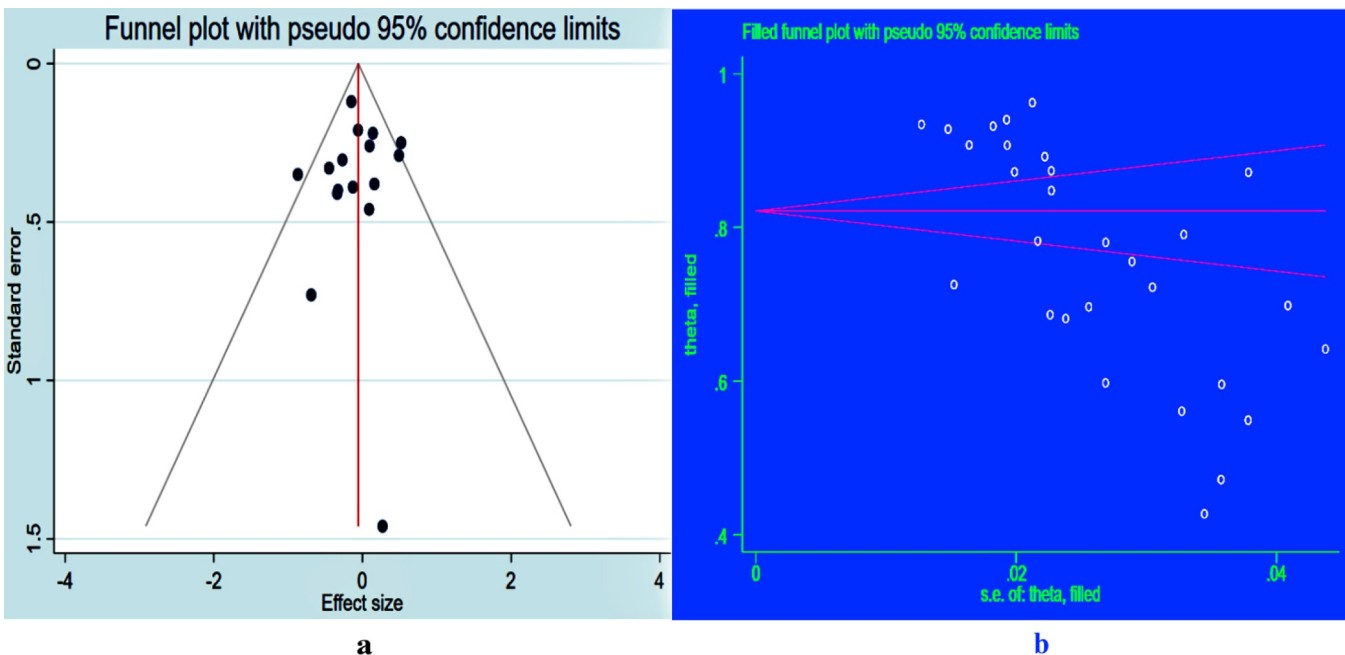

**Fig 6. Funnel plot showing the publication bias of level of adherence to ART among children living with HIV before adjustment "a" and after adjustment with trim and fill analysis "b" in Eastern and Southern Africa, 2024.**

## Discussion

This systematic review and meta-analysis was carried out to determine the overall pooled prevalence of adherence to ART and its associated factors among children living with HIV in ESA. In this study, the overall pooled prevalence of adherence to ART was 76.2% (95% CI: 71.4,

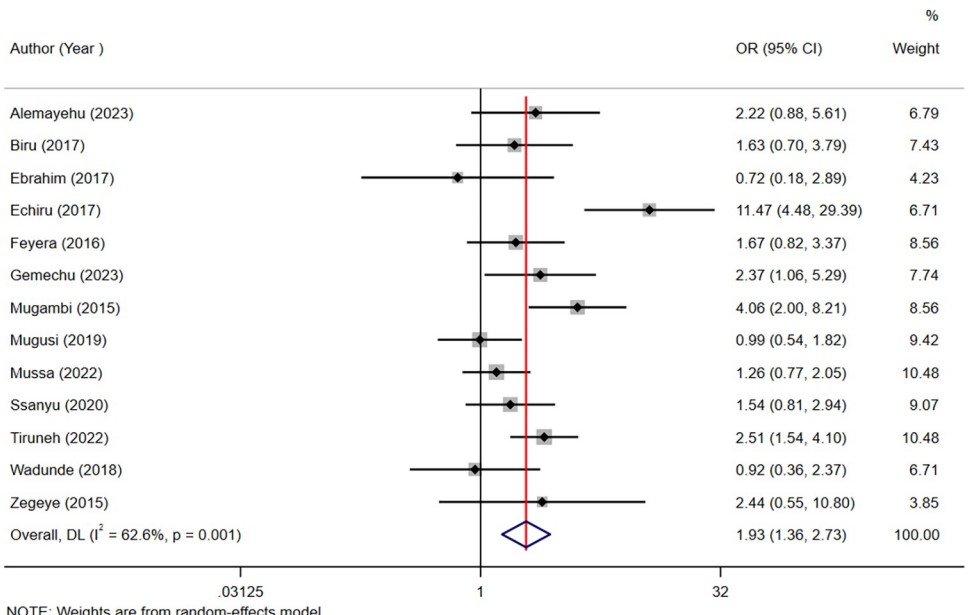

**Fig 7. Forest plots of the odds ratios with corresponding 95% CIs of studies on the association between the caregiver-child relationship and adherence to ART.**

**Table 2. The pooled effect size of the association between independent variables and adherence to ART based on a random effects model among children living with HIV.**

| Variables | Comparison | No of studies | Pooled OR/ AOR(95% CI) | Heterogeneity $I^2$ |
|---|---|---|---|---|
| Sex of the child [5, 12, 13, 15–17, 19, 22, 23, 25, 27, 29, 38, 39, 42, 43] | Female Vs. Male(Ref) | 16 | 0.95(0.80, 1.13) | 25.5% |
| Age of the Child [12, 14–16, 21] | 5–9 Vs. < 5 years | 5 | 0.94 (0.55, 1.62) | 53.7% |
| | 10–14 Vs. < 5 years | 5 | 1.12(0.56, 2.23) | 55.9% |
| Residence [5, 12, 16, 19–21] | Rural vs. Urban(Ref) | 6 | 0.81(0.41, 1.58 | 85.4% |
| Sex of caregiver [5, 12, 16, 17, 22, 23, 38, 39] | Male vs. Female (Ref) | 8 | 1.11 (0.77, 1.56) | 36.4% |
| Educational status of caregiver [5, 11, 12, 14–17, 20, 22, 25, 38, 39] | Primary education vs. No education (Ref) | 12 | 0.89(0.56, 1.36) | 66.8%, |
| | Secondary and above vs. No education (Ref) | 12 | 1.31(0.72, 2.38) | 79.8% |
| Caregiver-child relationship [5, 13, 14, 17–22, 29, 37–39] | Biological vs. non-biological(Ref) | 13 | **1.93(1.34, 2.73)*** | 62.6% |
| Knowledge of caregiver on adherence [11, 18, 20] | Poor vs. Good(Ref) | 3 | 0.56 (0.07, 4.81) | 93.7% |
| Distance from health facility [19, 22] | ≥10 km Vs. <10km(Ref) | 2 | 0.75 (0.41, 1.39) | 41.3% |
| WHO clinical stage [12, 25, 28, 29, 38, 39, 42] | stage III and IV vs. Stage I and II(Ref) | 7 | 1.09 (0.58, 2.05) | 82.9% |
| CD4 count [28, 39] | ≥ 500 Vs. <500(Ref) | 2 | 0.5(0.17, 1.46) | 56% |
| Viral Load [19, 20, 28] | ≥ 1000 Vs. <1000(Ref) | 3 | 1.53 (0.34, 6.99) | 88.4% |
| Disclosure status of HIV [11, 12, 14, 16, 18, 19, 21–23, 25, 29, 38, 39, 41] | Yes/No(Ref) | 14 | 1.47(0.93, 2.30) | 85.0% |
| Treatment stage [16, 19, 27] | First line vs. second line(Ref) | 3 | **2.7(1.39, 5.25)*** | 0 |
| Social support [21, 22, 28] | Yes/No(Ref) | 3 | **1.88(1.33, 2.66)*** | 0 |
| Side effect [5, 17, 22, 24, 27, 29] | Yes/No(Ref) | 6 | 0.62(0.33, 1.16) | 71.8% |

AOR: Adjusted Odds Ratio, Ref: Reference category, CI: Confidence Interval, *statistical significant

81.1; $I^2$ = 97.06%; P < 0.001). This overall pooled prevalence of adherence to ART is significantly lower than the WHO recommended threshold of ≥ 95% necessary to attain viral suppression [35, 36]. This finding is similar to those of studies conducted in China (77.6%) [48], Sub-Saharan Africa and North America (77%) [49], and SSA (72.9%) [50]. On the other hand, the result of this study is lower than when compared with studies conducted in Ethiopia (88.7% at 07 and 93.7% at 03 days prior to an interview [9] and South India (90.9%) [51], but higher than studies done elsewhere; worldwide (62%) [52], India (70%) [53], West Africa (42%) [54], and Ghana (70%) [30]. These variations could stem from differences in socio-demographic characteristics and adherence measurement methods. Since the assessment of some studies relied on caregiver reports, there's a possibility of adherence being overestimated due to a tendency to please the treatment provider and avoid criticism [9]. Additionally, the difference may arise from variations in the study location, methodologies employed, health-care delivery systems across different settings, and the presence of socio-cultural diversity [55]. Moreover, the discrepancies could be due to variations in socioeconomic status, variations in study time, as governments may have implemented intervention strategies to mitigate the issue, and differences in healthcare system policies. Furthermore, the variation could arise from differences in the ART regimen, the methods used to measure adherence to ART, and the clinical characteristics of the study participants. The study's finding of the overall pooled prevalence of adherence to ART of 76.2%, which significantly falls short of the WHO recommended threshold of over 95% necessary for effective viral suppression. This suboptimal adherence rate raises concerns about the increased risk of treatment failure and the potential development of drug-resistant strains, which could limit future treatment options and

exacerbate public health challenges. The lower adherence levels also suggest a higher risk of ongoing transmission due to unsuppressed viral loads, highlighting the urgent need for improved support systems that address barriers to adherence. Policymakers should focus on developing targeted interventions and investing in community-based programs, psychosocial support, and adherence counseling. Furthermore, continuous research and monitoring are essential to understanding adherence challenges and refining strategies to achieve better health outcomes and meet the WHO adherence targets. This indicates that urgent focus and actions are required to reduce the increased risk of ART drug resistance, opportunistic infections, hospitalization, illness, and death among HIV-infected children.

The subgroup analysis by study settings (country) showed that adherence to ART among children in ESA was inconsistent and ranged from 61 to 91%. The level of adherence to ART in South Africa was 91%, which is higher than studies done in Ethiopia (83%), Uganda (61%), Kenya (70%), Tanzania (67%), and South Sudan (70%); the lowest adherence was in Uganda. This variation may be due to beliefs regarding the benefits of HAART, religious practices, and the utilization of traditional medicine [56]. In addition, the discrepancy could arise from variations in healthcare systems and clinical settings, as well as differences in caregiver attitudes and awareness of HAART. Moreover, the heterogeneity of might be due to geographical variations among the studies that were included.

In addition, our subgroup analysis by year of publication showed a discrepancy. The pooled prevalence of studies published in 2020 and later 71% (95% CI: 63–78) were lower than the pooled prevalence of studies published before 2020 82% (95% CI: 76–87). The variations could be due to COVID-19 pandemic crisis. While the primary focus was on curbing the spread of COVID-19, UNAIDS notes that global efforts to combat HIV/AIDS has been disrupted by the pandemic [57]. Moreover, the variations might be due to differences in study periods across the primary studies that were included in the analysis. This showed that studies conducted in the later years had lower adherence to ART than earlier, emphasizing less focus and attention given recently.

There were pooled prevalence differences in the subgroup analysis by adherence measurements among the primary studies. The pooled prevalence of adherence among caregivers self-reported (reported questionnaire), and pill count were 76.7% (95% CI: 71.7, 81.8), and 71.6% (95% CI: 41.1, 102.1), respectively. This variation may arise from the fact that each of these methods has limitations and can provide different estimates of adherence. Caregiver self-reporting is the most common method of assessing adherence, although inaccuracy may result from imprecise or inconsistent questioning, patient forgetfulness, or the patient's desire to provide socially desirable answers. Nevertheless, questioning that is carefully structured, non-judgmental, and culturally appropriate may yield accurate information about adherence.

Children on first-line ART drugs in this study were almost three times more likely to adhere to ART than those on second-line ART drugs [AOR = 2.7 (95% CI: 1.39, 5.25)]. This finding was consistent with a study done in Ethiopia (AOR = 2.9) [9]. The possible justifications could be First-line ART drugs exhibit fewer side effects than second-line ART drugs [58]. If children are deemed unsuitable for first-line HAART due to drug resistance and/or treatment failure, it's possible that their adherence to HAART could be less likely [9]. If children exhibit adverse drug reactions, they can't take their ART drugs more likely, and as a result, they will adhere to ART less likely.

## Strengths and limitations of the study

This systematic review and meta-analysis is the first to combine the results of multiple studies conducted in ESA to show the pooled prevalence of adherence to ART and associated factors

among children living with HIV, providing stronger evidence in this HIV-high-burden region. The use of large sample size that spans the entire region (ESA) is also the strength of the study. We only included articles that were written in English, which may have caused us to overlook relevant articles written in Arabic, Swahili, French, or Portuguese. Since limited studies were conducted in some countries of the region, this restricted our ability to calculate a precise estimate of the prevalence of adherence in these countries.

## Conclusion and recommendation

The pooled prevalence of adherence to ART among children living with HIV in ESA was low. In the subgroup meta-analysis, variations among countries in the prevalence of adherence were observed, with the highest in South Africa and the lowest in Uganda. Besides, there was also variation based on year of publication, as the higher was observed in studies published before 2020 and the lower was observed in studies published in 2020 and later. Furthermore, the biological caregiver of the child, treatment with first-line ART drugs, and having social support were positively associated with adherence to ART among children living with HIV. Therefore, healthcare providers, adherence counselors, supporters, as well as governmental and non-governmental organizations, should emphasize a multi-component intervention approach to address the multifaceted challenges associated with adherence to ART, thereby improving treatment outcomes. Additionally, caregiver self-efficacy should rise for fostering children's dedication to consistently adhere to their ART medication regimen. Clinicians should prioritize the selection and utilization of regimens for individuals in this age group on robust first-line options and should carefully consider whether and how to introduce regimen changes in children. Furthermore, researchers should undertake implementation research that focuses on various domains of barriers to adherence to ART.

## Implications for policy and practice

The study's finding of low adherence to ART highlights an urgent need for targeted efforts to develop and implement comprehensive interventions. These should include patient-centered services, enhanced social support, home visits, and community-based assistance to improve adherence and achieve viral suppression. Clinicians, policymakers, and decision-makers can use these insights to refine ART services and bolster adherence support within existing healthcare systems. Effective implementation of these interventions is crucial for reducing drug resistance, decreasing new HIV infections, and mitigating HIV-related morbidity and mortality, particularly among children who are at higher risk due to poor adherence. By addressing these gaps, policymakers and the health system can improve health outcomes and advance the effectiveness of ART programs.

## Supporting information

**S1 Table. A 27-item Preferred Reporting Items for Systematic Reviews and Meta-Analyses (PRISMA 2020) checklist.**
(DOCX)

**S2 Table. Search strings used for a comprehensive search in databases.**
(DOCX)

**S3 Table. Quality appraisal results for included cross-sectional, cohort and experimental studies.**
(DOCX)

**S4 Table. Reasons for excluded primary studies.**
(DOCX)

**S1 Fig. Subgroup analysis by adherence measurements among HIV-infected children with ART in Eastern and Southern Africa from 2015 to 2024.**
(TIF)

## Acknowledgments

We would like to express our deepest gratitude and appreciation to the authors of the primary studies included in this systematic review and meta-analysis.

## Author Contributions

**Conceptualization:** Gebrie Getu Alemu, Bantie Getnet Yirsaw, Tigabu Kidie Tesfie, Getaneh Awoke Yismaw, Nebiyu Mekonnen Derseh.

**Data curation:** Gebrie Getu Alemu, Bantie Getnet Yirsaw, Habtamu Wagnew Abuhay, Nebiyu Mekonnen Derseh.

**Formal analysis:** Gebrie Getu Alemu, Tigabu Kidie Tesfie, Getaneh Awoke Yismaw, Meron Asmamaw Alemayehu, Muluken Chanie Agimas, Nebiyu Mekonnen Derseh.

**Investigation:** Gebrie Getu Alemu, Bantie Getnet Yirsaw, Tigabu Kidie Tesfie, Getaneh Awoke Yismaw, Habtamu Wagnew Abuhay, Meron Asmamaw Alemayehu, Muluken Chanie Agimas, Nebiyu Mekonnen Derseh.

**Methodology:** Gebrie Getu Alemu, Tigabu Kidie Tesfie, Getaneh Awoke Yismaw, Meron Asmamaw Alemayehu, Nebiyu Mekonnen Derseh.

**Project administration:** Gebrie Getu Alemu, Bantie Getnet Yirsaw, Habtamu Wagnew Abuhay, Muluken Chanie Agimas, Nebiyu Mekonnen Derseh.

**Resources:** Gebrie Getu Alemu.

**Software:** Gebrie Getu Alemu, Bantie Getnet Yirsaw, Tigabu Kidie Tesfie, Habtamu Wagnew Abuhay, Nebiyu Mekonnen Derseh.

**Supervision:** Gebrie Getu Alemu, Bantie Getnet Yirsaw, Getaneh Awoke Yismaw, Nebiyu Mekonnen Derseh.

**Validation:** Gebrie Getu Alemu, Nebiyu Mekonnen Derseh.

**Visualization:** Gebrie Getu Alemu, Bantie Getnet Yirsaw, Tigabu Kidie Tesfie, Getaneh Awoke Yismaw, Habtamu Wagnew Abuhay, Meron Asmamaw Alemayehu, Muluken Chanie Agimas, Nebiyu Mekonnen Derseh.

**Writing – original draft:** Gebrie Getu Alemu.

**Writing – review & editing:** Gebrie Getu Alemu, Meron Asmamaw Alemayehu, Muluken Chanie Agimas, Nebiyu Mekonnen Derseh.

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
