## [Decision Letter · Decision Letter 0]

17 Jul 2024

PONE-D-24-15644Adherence to Antiretroviral Therapy and its associated factors among children living with HIV in Eastern and Southern Africa: A Systematic Review and Meta-AnalysisPLOS ONE

Dear Dr. Alemu,

Thank you for submitting your manuscript to PLOS ONE. After careful consideration, we feel that it has merit but does not fully meet PLOS ONE’s publication criteria as it currently stands. Therefore, we invite you to submit a revised version of the manuscript that addresses the points raised during the review process.

In addition to the reviewer comment (appended), authors need to describe how they handle the diferent definitions of adherence among the studies which could impact on hetroginity. If not, indicate this as one of the limitation of this review.==============================

We look forward to receiving your revised manuscript.

Kind regards,

Yimam Getaneh Misganie, (PhD, PhD)

Academic Editor

PLOS ONE

Journal Requirements:

4. We note that this manuscript is a systematic review or meta-analysis; our author guidelines therefore require that you use PRISMA guidance to help improve reporting quality of this type of study. Please upload copies of the completed PRISMA checklist as Supporting Information with a file name “PRISMA checklist”.

Reviewers' comments:

Reviewer's Responses to Questions

**Comments to the Author**

1. Is the manuscript technically sound, and do the data support the conclusions?

Reviewer #1: Yes

2. Has the statistical analysis been performed appropriately and rigorously? 

Reviewer #1: I Don't Know

3. Have the authors made all data underlying the findings in their manuscript fully available?

Reviewer #1: Yes

4. Is the manuscript presented in an intelligible fashion and written in standard English?

Reviewer #1: No

5. Review Comments to the Author

Reviewer #1: The authors review an important topic in the field of paediatric HIV. They review various publications reporting on adherence in Eastern and Southern Africa and associated factors among children living with HIV. The findings reveal suboptimal adherence to ART in children.

The title and abstract are in alignment. The abstract summarises the main content of the paper.

Introduction seems to reference text from articles that is not representing the articles findings but rather the authors write up (from the background section) Eg line 62 reference (4); text is almost word for word.Also reference (5) line 63; may need to review the references.

The methods are appropriate

Outcome measure: various definitions of adherence were used; it would be interesting to know if prevalence rates varied among the different definitions. Was there any correlation between the reported adherence and viral suppression?

The discussion focuses on the findings and the associated factors of adherence.

6. PLOS authors have the option to publish the peer review history of their article (what does this mean?). If published, this will include your full peer review and any attached files.

Reviewer #1: No

---

## [Author Response · Author response to Decision Letter 0]

12 Aug 2024

Dear Editors and reviewer, Greetings.

We have submitted these authors' responses to peer-review comments and questions for a manuscript entitled: Adherence to antiretroviral therapy and its associated factors among children living with HIV in Eastern and Southern Africa: a systematic review and meta-analysis.

We have made the necessary corrections and responses to those comments raised by the academic editor and reviewer point by point, page by page, and line by line sequentially. We have submitted a rebuttal letter that responds to each point raised by the academic editor and reviewer with a file name of 'Response to Reviewers', a marked-up copy of our manuscript that highlights changes made to the original version labeled as 'Revised Manuscript with Track Changes' and an unmarked version of our revised paper without tracked changes labeled as 'Manuscript'. Thank you all for your contributions to this paper. If there will be any issues, we can contact further.

With regards! 

Gebrie Getu Alemu

gebryegetu27@gmail.com

University of Gondar, Ethiopia

PO. Box 196

Corresponding Author

---

## [Decision Letter · Decision Letter 1]

29 Aug 2024

PONE-D-24-15644R1Adherence to antiretroviral therapy and its associated factors among children living with HIV in Eastern and Southern Africa: a systematic review and meta-analysisPLOS ONE

Dear Dr. Alemu,

Thank you for submitting your manuscript to PLOS ONE. After careful consideration, we feel that it has merit but does not fully meet PLOS ONE’s publication criteria as it currently stands. Therefore, we invite you to submit a revised version of the manuscript that addresses the points raised during the review process.

We look forward to receiving your revised manuscript.

Kind regards,

Yimam Getaneh Misganie (PhD, PhD)

Academic Editor

PLOS ONE

Journal Requirements:

Reviewers' comments:

Reviewer's Responses to Questions

**Comments to the Author**

1. If the authors have adequately addressed your comments raised in a previous round of review and you feel that this manuscript is now acceptable for publication, you may indicate that here to bypass the “Comments to the Author” section, enter your conflict of interest statement in the “Confidential to Editor” section, and submit your "Accept" recommendation.

Reviewer #1 (New reviewer)

The authors review an important topic in the field of paediatric HIV. They review various publications reporting on adherence in Eastern and Southern Africa and associated factors among children living with HIV. The findings reveal suboptimal adherence to ART in children.

The title and abstract are in alignment. The abstract summarises the main content of the paper.

Introduction seems to reference text from articles that is not representing the articles findings but rather the authors write up (from the background section) Eg line 62 reference (4); text is almost word for word. Also reference (5) line 63; may need to review the references.

The methods are appropriate

Outcome measure: various definitions of adherence were used; it would be interesting to know if prevalence rates varied among the different definitions. Was there any correlation between the reported adherence and viral suppression?

The discussion focuses on the findings and the associated factors of adherence.

Reviewer #2: All comments have been addressed

2. Is the manuscript technically sound, and do the data support the conclusions?

Reviewer #2: Partly

3. Has the statistical analysis been performed appropriately and rigorously? 

Reviewer #2: Yes

4. Have the authors made all data underlying the findings in their manuscript fully available?

Reviewer #2: Yes

5. Is the manuscript presented in an intelligible fashion and written in standard English?

Reviewer #2: No

6. Review Comments to the Author

Reviewer #2: Greetings,

I am delighted to be invited to reviewing such interesting title. However, I think the study requires substantial revision including language editing.

The specific feedbacks are as follows:

1. Abstract (line 22): Please avoid the abbreviation, and write "antiretroviral therapy".

2. Methods session of the abstract (line 33): Please write "A random-effects DerSimonian-Laird model was used to compute the pooled prevalence of adherence to antiretroviral therapy among children living in Eastern and Southern Africa". The same is true in the main methods of the manuscript.

3. Methods session of the abstract (line 36): Please write "the potential sources of heterogeneity".

4. Methods session of the abstract (line 37): Please write "pooled adjusted odds ratio".

5. Results session of the abstract (line 40): Please write "the pooled prevalence of adherence to antiretroviral therapy among children living in Eastern and Southern Africa was 76.2% (95% CI: 71.4, 81.1) [I2 =41 97.07%, P < 0.001 and Q test (χ²) = 954.88, p-value < 0.001]".

6. Methods session of the abstract (line 43): "having social support" is vague! Is it to mean good social support? How social support was measured?

7. In the first paragraph of the introduction session, please state the burden of HIV, specifically to your target population (children).

8. Please also include the risk factors of adherence to ART in the introduction session.

9. Methods (line 104): Please avoid "by the University of York Centre for Reviews and Dissemination".

10. Search strategy session (line 121): What about citation tracking?

11. Why did you exclude those studies that have been conducted before January 2015?

12. Data extraction session (line 154): Please write "data were extracted". Data is plural, and datum is singular.

13. Data extraction session (line 155): Please write "Microsoft Excel spreadsheet".

14. Outcome measure session (line 160): Please write "Outcome measure". Under this session it is enough to mention the primary (prevalence of adherence to ART) and secondary outcome measure (factors associated with adherence to ART among…). And remove the rest.

15. Operational definition session (lines 171-177): Please remove the second paragraph (Caregiver :…).

16. Statistical analysis session (line 195): Please include the statistical value of Egger’s test.

17. Study characteristics session (line 241-251): Please also mention/include the range of response rates.

18. Table 1 did not correctly display the studys’ corresponding references. E.g. "Alemayehu, (2023)" was not correctly written, it must be written as "Alemayehu B (2023). The same is true for other references. And please put the order of authors based on the alphabetical order of the first authors.

19. It is better to add a reference index number to each study in the table (Table 1). Please also include the response rate of each included studies, and measuring tools that have been used to measure the outcome (adherence to ART).

20. Please write the "Data synthesis and statistical analysis" session after "Search outcome and study characteristics" session.

21. "Data synthesis and statistical analysis" session (line 197-217: Please mention/list the number of studies that have been involved to estimate the pooled prevalence of adherence to ART and the associated factors with their citations).

22. In the meta-analysis of factors associated with adherence (Figure 7), it seems that the authors only included studies reporting significant/positive association. However, it is more appropriate to include all studies that have investigated a given risk factor even if no significant association was found. Otherwise, the meta-analysis would lead to an overestimation of the true effect size.

23. Factors associated with adherence to ART session (line 327): Please write "among children living with HIV", and make it uniform throughout your manuscript.

24. During the subgroup analysis using study period, what was your rationality to say before 2020 and after 2020?

25. Meta-regression session (line 307-313): The content of the entire paragraph makes no sense and should be deleted. It is unclear what this analysis adds to the manuscript.

26. Who did you manage the publication bias?

27. Factors associated with adherence to ART among children living with HIV session (line 327-340): It is better to specify the total numbers of studies that showed a significant association with the outcome variable (adherence to ART among…) with their citations.

28. The discussion session needs more detailed narration with scientific rationalities supported with evidences/citations.

29. Conclusions and recommendations session (line 439): Please write "the pooled prevalence of adherence to ART among children living with HIV in ESA was low".

30. The recommendation session is non-specific and vague. Please make it specific and targeted with the identified risk factors affecting adherence to ART among children living with HIV.

7. PLOS authors have the option to publish the peer review history of their article (what does this mean?). If published, this will include your full peer review and any attached files.

Reviewer #2: **Yes: **Tigabu Munye Aytenew

---

## [Author Response · Author response to Decision Letter 1]

7 Sep 2024

Dear Editor,

We have submitted these authors' responses to peer-review comments and questions for a manuscript entitled: Adherence to antiretroviral therapy and its associated factors among children living with HIV in Eastern and Southern Africa: a systematic review and meta-analysis.

---

## [Decision Letter · Decision Letter 2]

16 Sep 2024

PONE-D-24-15644R2Adherence to antiretroviral therapy and its associated factors among children living with HIV in Eastern and Southern Africa: a systematic review and meta-analysisPLOS ONE

Dear Dr. Alemu,

Thank you for submitting your manuscript to PLOS ONE. After careful consideration, we feel that it has merit but does not fully meet PLOS ONE’s publication criteria as it currently stands. Therefore, we invite you to submit a revised version of the manuscript that addresses the points raised during the review process.

We look forward to receiving your revised manuscript.

Kind regards,

Yimam Getaneh (PhD, PhD)

Academic Editor

PLOS ONE

Journal Requirements:

Reviewer's Comment:

**Comments to the Author**

1. If the authors have adequately addressed your comments raised in a previous round of review and you feel that this manuscript is now acceptable for publication, you may indicate that here to bypass the “Comments to the Author” section, enter your conflict of interest statement in the “Confidential to Editor” section, and submit your "Accept" recommendation.

Reviewer #2: All comments have been addressed

2. Is the manuscript technically sound, and do the data support the conclusions?

Reviewer #2: Yes

3. Has the statistical analysis been performed appropriately and rigorously? 

Reviewer #2: No

4. Have the authors made all data underlying the findings in their manuscript fully available?

Reviewer #2: Yes

5. Is the manuscript presented in an intelligible fashion and written in standard English?

Reviewer #2: Yes

6. Review Comments to the Author

Reviewer #2: I'm pleased to see the second revised manuscript where the major issues have been addressed. The specific feedbacks are also as follows:

1. Abstract (line 31-32): Please replace with "Data were extracted and analyzed using Microsoft Excel spreadsheet and STATA version 17 software, respectively"

2. Abstract (line 35): Since you have already removed the meta-regression, please remove the "Meta-regression".

3. Abstract (line 49-52): The recommendation is not specific/in line with your findings. Please revise it.

4. Your framework will not be CoCoPoP, rather it must be of the PECO, consists of population (P), exposure (E), context (C), and outcome (O).

5. Line 140: "in children < 15 years on ART" shall be replaced with "among children aged <15 years" and which one is a representative for your target population (≤14 years or <15 years)?

6. Line 150: Are quality appraisal and risk of bias assessment similar?

7. Line 172-179: You can use a single paragraph using conjunctions.

8. Please take the figure legends at the end of your reference.

9. Line 224-225: Please replace with "The extracted data were exported to STATA version 17 software for statistical for analysis"

10. Line 242; Please remove "meta-regression"

11. Line 252: The pooled prevalence of adherence to ART was 76.2% (95% CI: 71.4, 81.1). However, Figure 2 stated that the pooled prevalence was 0.76 (95% CI: 0.71, 0.81), which is statistically different with your report, indicating your data extraction was incorrect. It also contradicts with the study characteristics (line 250-251). Please correct it again. The same is true for other figures.

12. Line 326: Replace with …"the overall pooled prevalence of adherence to ART"… and make it consistent.

13. Line 429-433: Please put each abbreviation on a new line.

14. Line 434: Please remove it.

15. Line 435-436: Please put it at the end of the methods section in the manuscript.

7. PLOS authors have the option to publish the peer review history of their article (what does this mean?). If published, this will include your full peer review and any attached files.

Reviewer #2: **Yes: **Tigabu Munye Aytenew

---

## [Author Response · Author response to Decision Letter 2]

7 Oct 2024

Dear Editors and reviewer, Greetings.

We have submitted the authors' responses to peer-review comments and questions for a manuscript entitled: Adherence to antiretroviral therapy and its associated factors among children living with HIV in Eastern and Southern Africa: a systematic review and meta-analysis.

We have made the necessary corrections and responses to those comments raised by the academic editor and reviewer point by point, page by page, and line by line sequentially. We have submitted a rebuttal letter that responds to each point raised by the academic editor and reviewer with a file name of 'Response to Reviewers', a marked-up copy of our manuscript that highlights changes made to the original version labeled as 'Revised Manuscript with Track Changes' and an unmarked version of our revised paper without tracked changes labeled as 'Manuscript'. Thank you all for your contributions to this paper. If there will be any issues, we can contact further.

With regards! 

Gebrie Getu Alemu

gebryegetu27@gmail.com

University of Gondar, Ethiopia

PO. Box 196

Corresponding Author

---

## [Editor Report · Decision Letter 3]

9 Oct 2024

Adherence to antiretroviral therapy and its associated factors among children living with HIV in Eastern and Southern Africa: a systematic review and meta-analysis

PONE-D-24-15644R3

Dear Dr. Alemu,

We’re pleased to inform you that your manuscript has been judged scientifically suitable for publication and will be formally accepted for publication once it meets all outstanding technical requirements.

These technical requirements may also include revising the manuscript according to journal guidelines, such as adjusting the layout, ensuring consistent font sizes, using specific section titles in the abstract (e.g., "Background" instead of "Introduction"), and adhering to the word count limit. 

Kind regards,

Yimam Getaneh (MSc, PhD, PhD)

Academic Editor

PLOS ONE

---

## [Editor Report · Acceptance letter]

1 Nov 2024

PONE-D-24-15644R3 

PLOS ONE

Dear Dr. Alemu, 

I'm pleased to inform you that your manuscript has been deemed suitable for publication in PLOS ONE. Congratulations! Your manuscript is now being handed over to our production team.

Kind regards, 

on behalf of

Dr. Yimam Getaneh Misganie 

Academic Editor

PLOS ONE